# Evaluation of Serological Methods and a New Real-Time Nested PCR for Small Ruminant Lentiviruses

**DOI:** 10.3390/pathogens11020129

**Published:** 2022-01-21

**Authors:** Jessica Schaer, Zeljko Cvetnic, Tomislav Sukalic, Sven Dörig, Martin Grisiger, Carmen Iscaro, Francesco Feliziani, Folke Pfeifer, Francesco Origgi, Reto Giacomo Zanoni, Carlos Eduardo Abril

**Affiliations:** 1Institute of Virology and Immunology IVI, in Cooperation with the Vetsuisse-Faculty of the University of Bern, 3012 Bern, Switzerland; je.schaer@gmail.com (J.S.); reto.zanoni@vetsuisse.unibe.ch (R.G.Z.); 2Regional Veterinary Department Križevci, Croatian Veterinary Institute, Zakmandijeva 10, 48260 Križevci, Croatia; cvetnic@veinst.hr (Z.C.); sukalic.vzk@veinst.hr (T.S.); 3Beratungs-und Gesundheitsdienst für Kleinwiederkäuer (BGK/SSPR), 3362 Niederoenz, Switzerland; Sven.Doerig@caprovis.ch; 4Veterinaerdienst der Urkantone, 6440 Brunnen, Switzerland; Martin.Grisiger@laburk.ch; 5National Reference Laboratory for Ruminant Retroviruses, Istituto Zooprofilattico, Sperimentale dell’Umbria e delle Marche Togo Rosati, 06126 Perugia, Italy; c.iscaro@izsum.it (C.I.); f.feliziani@izsum.it (F.F.); 6Tierseuchenkasse/Tiergesundheitsdienst Sachsen-Anhalt, 39116 Magdeburg, Germany; f.pfeifer@tskst.de; 7Institute of Veterinary Pathology, Vetsuisse-Faculty of the University of Bern, 3012 Bern, Switzerland; francesco.origgi@vetsuisse.unibe.ch

**Keywords:** small ruminant lentivirus, Maedi-Visna, caprine arthritis-encephalitis, diagnosis, serological methods, molecular methods, ELISA, nested real-time PCR

## Abstract

Small ruminant lentiviruses (SRLVs), i.e., CAEV and MVV, cause insidious infections with life-long persistence and a slowly progressive disease, impairing both animal welfare and productivity in affected herds. The complex diagnosis of SRLVs currently combines serological methods including whole-virus and peptide-based ELISAs and Immunoblot. To improve the current diagnostic protocol, we analyzed 290 sera of animals originating from different European countries in parallel with three commercial screening ELISAs, Immunoblot as a confirmatory assay and five SU5 peptide ELISAs for genotype differentiation. A newly developed nested real-time PCR was carried out for the detection and genotype differentiation of the virus. Using a heat-map display of the combined results, the drawbacks of the current techniques were graphically visualized and quantified. The immunoblot and the SU5-ELISAs exhibited either unsatisfactory sensitivity or insufficient reliability in the differentiation of the causative viral genotype, respectively. The new truth standard was the concordance of the results of two out of three screening ELISAs and the PCR results for serologically false negative samples along with genotype differentiation. Whole-virus antigen-based ELISA showed the highest sensitivity (92.2%) and specificity (98.9%) among the screening tests, whereas PCR exhibited a sensitivity of 75%.

## 1. Introduction

Maedi-Visna virus (MVV) and caprine arthritis-encephalitis virus (CAEV) belong to the family Retroviridae, and the genus Lentivirus. They are members of a heterogeneous group called the small ruminant lentiviruses (SRLVs) comprising five different genotypes (A, B, C, D and E), infecting goats and sheep, causing cross- and superinfections [1,2,3,4,5]. MVV-like and CAEV-like strains belong to genotype A and B, respectively, with a worldwide distribution. The transmission routes for SRLVs are principally vertically through colostrum or milk but also horizontally through respiratory secretions, favored by overcrowding [6]. SRLV infections persist for life, whereby the mounted specific immune response does not protect against disease or superinfection [7]. Only one-third of the infected animals show clinical signs [8,9]. Still, SRLVs have a significant economic impact on animal production and affect animal welfare [10]. The presence of specific antibodies is useful as an indicator of an infection and therefore, eradication and surveillance programs are based on the serological detection of infected animals. The most frequently used assays are the agar gel immunodiffusion (AGID) test and the enzyme-linked immunosorbent assay (ELISA) (OIE Terrestrial Manual, https://www.oie.int/fileadmin/Home/eng/Health_standards/tahm/3.07.02_CAE_MV.pdf, accessed on 6 December 2021). For the AGID test, the two major SRLV antigens used routinely are the glycoprotein 135 (gp135) and the capsid protein (p28), respectively. The specificity of the AGID test is high, but its sensitivity has been reported as being low [11,12,13]. Still, a recent study carried out on Belgian sheep and goats [13] reported a sensitivity and specificity of 100% by combining the results of two AGID kits. Several ELISA tests have been developed and described in the literature using whole virus preparations, recombinant proteins or synthetic peptides, mostly designed as indirect but also as competitive assays (according to the OIE Terrestrial Manual). The high sensitivity, efficient handling and ease of interpretation of the results are the major advantages of the ELISA compared to AGID tests. For this reason, the ELISA tests are preferred for surveillance and eradication programs. A major drawback of serology is the high genetic variability of the SRLVs, translating into high antigenic diversity [14,15]. Therefore, the serological diagnosis of SRLV remains quite challenging [7,12,16] in the absence of a “gold standard” [17]. The high genomic variability also complicates the design of tools for the molecular detection of SRLV. Furthermore, apart from low viral load, no free virus is detectable in the blood of naturally infected animals using PCR methods. Therefore, these methods mainly target the provirus in peripheral blood leukocytes [18,19].

The current SRLV diagnostic procedure in Switzerland is based on a protocol consisting of an array of four sequentially used serological tests [20] performed on individual seropositive—not herd—samples. The sampling follows field criteria, e.g., animal purchase, animal transport or clinical suspicion. The serum samples are initially screened with the IDEXX CAEV/MVV Total Ab Test (Idexx Laboratories, Liebefeld, Switzerland), a whole-virus antigen-based indirect ELISA [21] and the Small Ruminant Lentivirus Antibody Test Kit VMRD (VMRD, Pullman, WA, USA), which is a gp135-based competitive ELISA [22]. Subsequently, the immunoblot based on whole virus antigen is performed as a confirmatory assay for seropositive samples [23] followed by the SU5-ELISA set as a serotyping differentiation test into the subtypes A1, A3, A4, B1 and B2 [24]. As far as the voluntary MVV control program in sheep is concerned, the sera are screened initially for SRLV antibodies using the Eradikit^®^ SRLV Screening Kit (In3 diagnostic, Via B.S. Valfrè, 18, 10,121 Torino, Italia) before confirmation as described above. This thorough diagnostic procedure is expensive, laborious and time-consuming, and still quite frequently leads to inconclusive or contradictory results. Therefore, the aim of this work was to evaluate more closely the performance of the single and combined tests used, with the goal of optimizing the overall procedure. Additionally, we designed a nested real-time PCR targeting highly conserved genomic regions [25] and evaluated the benefit of the inclusion of this tool in the current diagnostic strategy.

## 2. Results

### 2.1. Determination of SRLV True Positive Standard

The results obtained for the 290 samples used in the serological screening, confirmation and genotype differentiation tests and the nested real-time PCR are visualized in a graphical heatmap format in Table 1. The colors reflect the intensity of each serological reaction, with values ranging from dark to light blue to white (negative) and from light to dark red (positive). In general, positive serological reactions were seen in all flocks investigated, indicating the presence of SRLV infections. However, overall, remarkable discordance between the results obtained by different serological tests was evident(*), avoiding the recognition of clear serological reactivity patterns within flocks. The highest agreement among the screening and confirmatory test results was seen in the samples of flocks 2, 3 and 5, whereas flock 10 exhibited discordance among the results of the screening tests as well as among screening and confirmation (Table 1).

In order to evaluate the performance of each test in the absence of a generally accepted gold standard, the SRLV true positive status was identified as a composite reference standard as recommended by others [13,16,26,27,28]. For that purpose, the composite concordance of the serological results across the different screening tests was analyzed, as shown in the Venn diagram (Figure 1). A total of 137 samples were positive with all three commercial tests, and therefore were considered as serologically true positives. The IDEXX-, VMRD- and ERADIKIT ELISA tests detected 178, 177 and 170 positive samples, respectively. The results of the Immunoblot (IB) formerly used as a serological and real-time PCR as an additional truth standard were included for comparative purposes (Figure 2). In the non-overlapping areas, one of four single-positive IDEXX ELISA samples showed a positive IB result, and three of them showed a positive real-time PCR result. None out of seven single-positive VMRD ELISA samples were positive in IB; however, three were positive by PCR. Finally, none out of nine single-positive ERADIKIT ELISA samples were positive in IB, and only one was positive by PCR. The proportion of real-time PCR positive samples was highest in single-positive IDEXX ELISA samples. In the overlapping regions between the IDEXX ELISA with either the ERADIKIT ELISA or the VMRD ELISA, more IB and real-time PCR positive results were observed than in the overlapping region between the ERADIKIT- and the VMRD ELISA. Samples’ ELISA reactivity tended to be higher in the overlapping areas of the Venn diagram than that of samples in the non-overlapping regions (data not shown). Based on these results, we arbitrarily defined the condition of a positive reaction with at least two screening ELISAs as a composite standard rule for serological truth, complemented with real-time PCR as an additional, independent truth standard.

### 2.2. Performance of the Serological Tests Based on the Composite Truth Standard vs. Real-Time PCR

Sensitivity and specificity were calculated by crossing the results of each test with the composite truth standard (Table 2). The IDEXX ELISA exhibited a sensitivity of 92.2%, a specificity of 98.9%, with a positive (PPV) and a negative (NPV) predictive value of 99,4% and 85.9%, respectively, whereas the VMRD ELISA’s sensitivity, specificity, PPV and NPV were 90.1%, 95.7%, 97.7% and 82.2%, respectively. The lowest performance was observed with the ERADIKIT ELISA with a sensitivity of 84.4%, a specificity of 91.3%, a PPV of 95.3% and an NPV of 73.7%. The distribution of the results of the commercial ELISA tests is shown as dot plots in Figure 2, revealing the IDEXX ELISA as the clearest discriminant between the expected negative and positive samples, followed by VMRD and the ERADIKIT ELISA tests. The immunoblot, which was formerly used as a confirmatory test, showed a low sensitivity of 59.2%, a specificity of 92.4%, a PPV of 94.2% and an NPV of 52.2%. The performance of the nested real-time PCR as the new, independent truth standard exceeded that of the immunoblot, with a sensitivity of 75.5%, a specificity and PPV of 100%, and an NPV of 66.2% (Table 3).

Comparing the results regarding the animal host or viral species (Table 4 and Table 5), the immunoblot exhibited more false negative samples in sheep (50%) than in goats (5.9%, Table 4. Still, from the view of the viral species, the results of the immunoblot show a clear tendency toward more false negative results in samples identified as MVV than in samples identified as CAEV by PCR (Table 5). No significance tests were applied due to the small sample size of goats.

### 2.3. Serological and Molecular Differentiation between MVV and CAEV Infections

The SU5 ELISA had been used to date to differentiate between MVV and CAEV infections in seropositive samples. According to the composite truth standard, 24 samples were categorized as false positive. Due to its use as a differentiation test, we restricted the evaluation of SU5 ELISA to the serological differentiation between MVV and CAEV. In total, 151 out of 184 serologically positive samples could be characterized as MVV- or CAEV-positive. Samples without positive SU5 test results were classified as “SRLV-positive” (Table 1). Using the nested real-time PCR for differentiation, a total of 147 samples were classified either as MVV or CAEV, whereas in six samples, coinfection with both MVV and CAEV was detected. In a total of 57 real-time PCR positive samples, the classification was confirmed by sequencing (Figure 3). The differentiation between MVV and CAEV by real-time PCR was 100% concordant with the phylogenetic analysis of the sequences using both NCBI BLAST (basic local alignment search tool [29]) and phylogenetic analysis. To evaluate the agreement between the SU5 ELISA and the real-time PCR, the results of 116 samples available for both methods were compared. The real-time PCR results were considered as true on the basis of the sequence confirmation. Fifty-nine and 51 samples were differentiated as CAEV and MVV, respectively, by real-time PCR (Figure 2d). Good concordance between the real-time PCR and SU5 ELISA results (48 out of 51 samples) was observed in the samples classified as MVV by real-time PCR (Figure 2d). However, 19 out of 59 samples classified as CAEV were misclassified as MVV by SU5-ELISA. Furthermore, the real-time PCR was able to detect coinfection with both virus types in six samples (Table 1). 

### 2.4. Phylogenetic Analysis of Sequences

The genetic relatedness among the SRLV was analyzed using a 200 bp-long LTR-gag sequence fragment located within the real-time PCR target sequence. A total of 57 sequences supplemented with 14 closely related sequences retrieved from GenBank were analyzed phylogenetically (Figure 3). Two main clusters with clear separation between genotype A and genotype B according to the reference sequences were observed in the phylogenetic tree. Thus, the clusters were denominated as genotype A and genotype B (Figure 3). Moreover, as mentioned above, the classification as MVV or CAEV by real-time PCR was in complete agreement with the genotype classification in the phylogenetic tree.

The genetic relatedness of the sequences characterized was higher within flocks or within given geographic regions. The sequences from Italian samples in the genotype B cluster, split into two different clades closely related to Italian sequences previously characterized as subtype B2 (MG554402) and B3 (JF502417) strains [30], respectively. Additionally, four Italian samples were placed as separate groups in the genotype A cluster. These sequences were closely related to representative sequences of the subtypes A19 (MH374287) and A8 (MH374284), respectively, which similarly originated from Italy. The sequences of Croatian samples were also present in both the genotype A and B clusters of the tree. Two sequences located in the genotype B were found to be related to the subtype B1 reference sequence of the strain CAEV-Co (M33677.1) and three to the representative subtype B1 sequence MG554410. Sequences of genotype A were located in three different clades and were closely related to the representative sequences of the subtypes A18 (MG554409) and A4 (AY445885). Interestingly, due to sequence divergence, two distant clades were related to the same subtype A18 (MG554409) sequence. This divergence was also reflected in the percentages of identity among the samples of the individual clades (data not shown) and additionally in the different ranges of identity observed between the two clades and the subtype A18 (MG554409) sequence (Figure 4). Given the most commonly accepted SRLV subtyping method, it would be interesting to sequence the gag gene of these samples, which was beyond the focus of this work.

The 17 German samples were sent for diagnostic and epidemiological purposes. The only two positive goat samples were serologically positive repetitively in a German SRLV surveillance program. Our serological and genotyping results were in complete concordance with those of the Friedrich-Loeffler Institut (FLI, personal communication). The sequence of the German sample classified as genotype A was closely related to the subtype A2 sequence (MT993908), and that classified as genotype B was linked to the sequence of the subtype B1 CAEV-Co strain.

The Swiss samples were almost exclusively from sheep with only two samples from goats, which tested negative with all the protocols used. Sequence analysis revealed a balanced distribution of the Swiss sheep samples within the genotype A and B clusters. Samples with genotype B sequences were exclusively found in the clade with the subtype B1 CAEV-Co reference strain, whereas the genotype A samples were located in four different clades. The closest relatives of these four clades were sequences of the subtypes A18 (MG554409), A4 (AY445885) and A2 (MT993908), respectively (Figure 3).

## 3. Discussion

The diagnosis of SRLV infections is based on the serological detection of infected animals [3,12,31,32,33,34,35,36,37]. Due to low viral load and genomic heterogeneity, direct detection of the virus in blood is considered less efficient [3,12,16,32,33,38,39,40,41,42,43,44,45], though not negligible in view of delayed seroconversion [18,46,47,48,49,50,51,52,53]. In this work, using a panel of 290 sera of animals originating from different European countries, we aimed to improve and simplify the quite complex diagnostic protocol established in our diagnostic laboratory consisting of several serological tests run in parallel. Heat-mapping of the results from three commercial screening ELISAs, Immunoblot as a confirmatory assay and five SU5 peptide ELISAs for genotype differentiation exhibited a quite complex and challenging pattern of reactivity with a number of inconsistent results. Therefore, it was crucial to define, based on the given serological results, an overall truth standard. We based this decision on the well-accepted concept of using a “composite standard” [13,16,26,27,28] defined by the three screening ELISAs used. A useful complementation of this purely serological approach was the addition of a newly developed, highly sensitive PCR consisting of a primary round of a conventional PCR followed by a nested real-time PCR with the ability to differentiate between CAEV and MVV due to reliably discriminating probes with 100% specificity. The nested approach using serial dilutions of plasmids containing the target sequences exhibited at least 10-fold higher sensitivity compared to simple real-time PCR (data not shown).

This composite truth for determining the condition of infection status of a given animal allowed us to re-evaluate the current diagnostic procedure performed in our laboratory. IDEXX ELISA showed the highest sensitivity and specificity out of the three screening tests and the combination of IDEXX ELISA and VMRD ELISA detected the most “true positives” compared to the combinations of IDEXX-/ERADIKIT ELISA and VMRD-/ERADIKIT ELISA. The Immunoblot based on CAEV whole virus used for confirmation exhibited an unsatisfactory sensitivity of 59.2%. This is in accordance with its tendency toward inconclusive results due to an unexplained nonspecific reactivity of most sera with the capsid protein p25 of CAEV seen over the years (data not shown). Furthermore, we show a weaker reactivity of the immunoblot in the case of MVV compared to CAEV infections (Table 5), which does not appear to be due to either the host species or the antigen used (not shown). Given this rather poor performance, the Immunoblot is a quite laborious, demanding and antigen-consuming confirmatory assay, and we would recommend its exclusion from the diagnostic procedure.

PCR exhibited a nearly equal overall sensitivity (75.5%) compared to the SU5-ELISA set (80.7%), combined with 100% specificity and consequent unambiguous differentiation between the infection source (CAEV or MVV). Therefore, it seems appropriate to exclude the labor-intensive and costly SU5 peptide ELISA approach consisting of five ELISAs in favor of the PCR protocol as a differentiation tool. The application of the PCR in the diagnostic procedure has additional advantages regarding the possibility of detecting seronegative animals and sequencing the amplification products for molecular epidemiological analysis. Furthermore, the chance to achieve specific amplification in a seropositive sample might be significantly higher, apparently depending on the sample quality according to our laboratory experience (up to 85%, not shown). This point combined with the determination of the analytical sensitivity of the nested real-time PCR should be elucidated in further applied studies.

As also shown by others [12,43,54,55,56,57,58,59,60,61,62], the resulting LTR sequence fragments of this work allowed quite a robust and plausible phylogentic reconstruction of the taxonomy of lentiviruses detected in the samples originating from a number of Western European countries. Sequencing of the whole genome or other fragments thereof to exclude homologous recombination could be a useful complementation for future work.

In conclusion, given these results, the diagnosis of SRLV infection by a reference laboratory could actually be reduced to the combined application of well-established screening ELISAs complemented with the PCR described here as a differentiation tool regarding the source of infection, i.e., CAEV or MVV, respectively. Finally, with only minor loss of overall performance (not shown), the screening could be reduced to two ELISA kits, e.g., IDEXX ELISA combined with VMRD ELISA and PCR.

## 4. Materials and Methods

### 4.1. Samples

EDTA-anticoagulated whole blood samples from 290 animals originating from different countries including Croatia, Germany, Italy and Switzerland were analyzed (Table 1). Animals were sampled during 2019 and 2020 for serological and molecular diagnosis of SRLV infections. All flocks tested positive for SRLV infection previously. Croatian samples consisted of 50 samples from goats of 5 flocks (10 each). The Italian samples consisted of 51 samples from sheep of 3 flocks located in south of Italy (11, 20 and 20, respectively). German samples consisted of 17 samples from goats of one flock. The samples originating from Switzerland were sent to our laboratory for SRLV diagnosis consisting of 170 sheep and 2 goat samples from 18 different flocks with one or more samples per flock (Table 6).

### 4.2. Serology

Blood samples were analyzed for the presence of SRLV antibodies using three commercial ELISA kits in parallel, namely the IDEXX ELISA (IDEXX CAEV/MVV Total Ab Test (Idexx Laboratories, Liebefeld, Switzerland; whole-virus antigen based indirect ELISA), VMRD ELISA (Small Ruminant Lentivirus Antibody Test Kit-VMRD, Pullman, WA, USA; genotype B gp135 competitive ELISA) and the ERADIKIT ELISA (Eradikit^®^ SRLV Screening Kit-In3Diagnostic, Torino, Italy; *gag* and *env* peptide based indirect ELISA), followed by immunoblotting and an in-house SU5 peptide ELISA test [24]. The cut-off values for the commercial ELISA tests were determined according to the manufacturers’ guidelines.

The Immunoblot confirmation was based on the detection of the viral capsid (p25), matrix (p15) and nucleocapsid (p18) proteins [23]. Reactive bands were visually evaluated and scored from 1 to 3 according to the band intensity. Samples showing staining of at least two bands or single bands scored as 2 or more were considered as positive.

The SU5 peptide ELISA was implemented for genotype differentiation. This test consisted of five ELISA tests containing synthetic peptides of the immunodominant SU5 region of the SRLV subgenotypes A1, A3, A4, B1 and B2. Samples with an OD value equal to or over 30% were considered as positive.

### 4.3. Multiple Sequence Alignments and Design of Primers and Probes

Primers and probes for the nested real-time PCR system are shown in Table 7. Whole genome sequence alignment comprising 52 published SRLV sequences retrieved from the National Centre for Biotechnology Information (NCBI) was used for the selection of conserved genomic regions and the design of the nested real-time PCR system. The Geneious prime software version 2020 (https://www.geneious.com, accessed on 6 December 2021) was used for alignment and the Primer Express^®^ Software Version 3.0.1 (Applied Biosystems) was used for the design of primers and probes.

The forward primers (outer primer–F) of the first amplification step were designed to anneal to the highly conserved lentiviral RNAt^lys^ primer-binding site (PBS) located at the leader region of the lentiviral genome. The similarly conserved annealing region of the reverse primers (outer primer–R) is located at 87 and 108 bp downstream of the predicted ATG gag initiation codon of the genotype A (GenBank Acc. No M60610) and B (GenBank Acc. No. M33677), respectively (outer primer–R). The forward primers for the genotype-specific real-time PCRs are located in a previously described conserved region encompassing a stem-loop structure that contains the dimer initiation site (DIS), just upstream of the major splice donor (MSD) of the small ruminant lentiviruses [63,64]. The reverse primers and probes are located immediately downstream of the predicted gag start codon. The target sequences of the real-time PCR reverse primers and probes and that of the outer reverse primers used were described previously [65,66]. A schematic diagram showing the localization of the nested real-time PCR target sequences is shown in Figure 3.

### 4.4. DNA Extraction and Two-Step Nested Real-Time PCR System

Seven hundred and fifty microliters of EDTA blood were treated with ammonium chloride/Tris buffer (0.14 M NH4Cl, 0.17 M Tris, pH 7.2) to obtain the buffy coats [67]. Buffy coats were stored at −20 °C or used directly for DNA extraction. DNA extraction was performed using the Qiagen DNeasy^®^ Blood & Tissue kit (Qiagen, Hilden, Germany) and finally the DNA was eluted in 100 µL of elution buffer according to the manufacturer’s protocol.

The nested real-time PCR was carried out in two successive amplification steps. The first step PCR reaction consisting of a conventional qualitative PCR contained 12.5 µL of Hotstar Taq Master Mix (HotStarTaq DNA Polymerase kit, Qiagen GmbH, Hilden, Germany), 300 nM of each primer and 5 µL of the extracted DNA. Amplification started with the activation of the polymerase at 95 °C for 15 min, followed by 40 cycles at 95 °C for 20 s, and at 60 °C for 30 s. All products of the first PCR were then tested in parallel with the genotype-specific real-time PCRs (second step of the nested real-time PCR protocol) for a sensitive detection and discrimination between genotypes A and B.

Real-time PCR reactions contained 12.5 µL TaqMan™ Universal PCR Master Mix (Applied Biosystems, Life Technologies), 900 nM of each primer, 200 nM probes and 5 µL of the PCR product of the first step. Amplification profiles consisted of a hold stage of 20 s at 95 °C and a PCR stage of 40 cycles at 95 °C for 15 s and 60° C for 1 min. Thermal cycling was performed with a 7300 Real-Time PCR System (Applied Biosystems, Life Technologies).

### 4.5. Sequencing and Sequence Analysis

Samples with positive nested real-time PCR results were submitted for sequencing of the real-time PCR target region (Microsynth AG, Balgach. Switzerland). Sequences obtained were aligned using the clustal omega algorithm implemented in Geneious Prime software (Geneious Prime 2020.2.4). The phylogeny was inferred using the maximum likelihood method and the Tamura–Nei substitution model. Phylogenetic analyses were conducted using MEGA X [68]. The sequences from PCR products were deposited in GenBank under the following accession numbers: OL456240, OL456241 and OL449029 to OL449084.

### 4.6. Data Analysis

Sensitivity and specificity of serological tests and PCR were calculated using Single Sample Binary Diagnostic Tests by the software NCSS (NCSS Statistical Software (2019). NCSS, LLC. Kaysville, UT, USA, ncss.com/software/ncss, accessed on 6 December 2021). 

## Figures and Tables

**Figure 1 pathogens-11-00129-f001:**
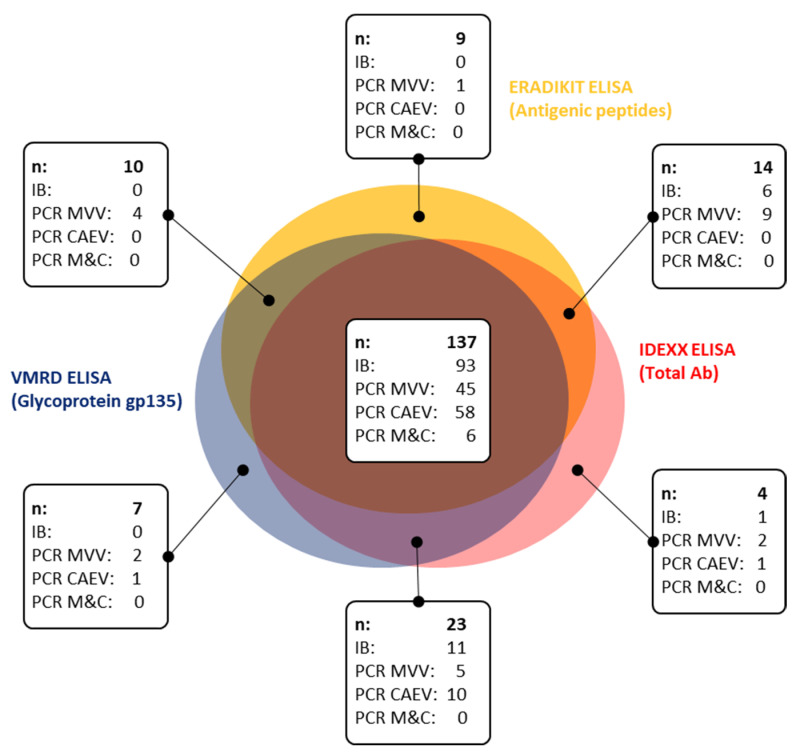
Venn diagram showing the concordance between serological results. Number of samples (*n*), results of the immunoblot test (IB), and real-time PCR results (PCR MVV: positive for Maedi-Visna virus; PCR CAEV: positive for the caprine arthritis-encephalitis virus; and PCR M&C: samples that were positive for both virus types) are indicated in the boxes.

**Figure 2 pathogens-11-00129-f002:**
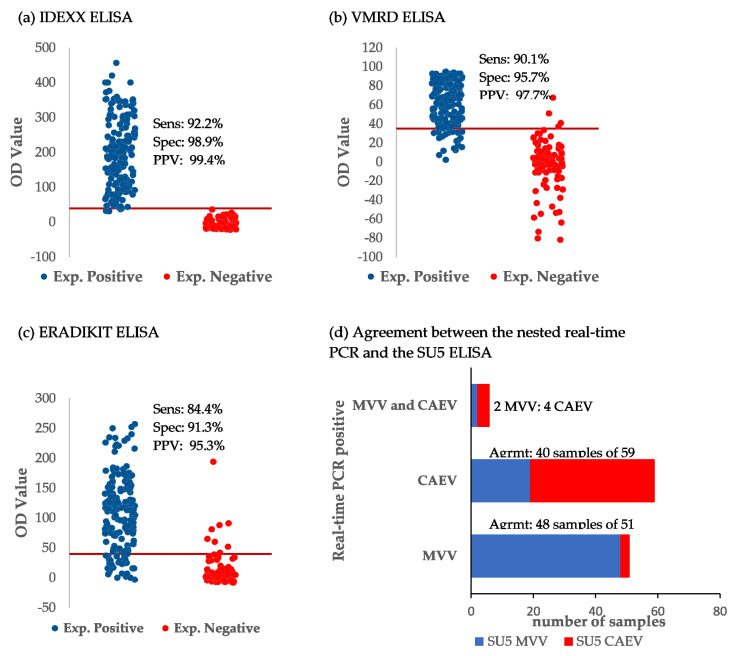
Jitter plots (**a**–**c**) and histogram (**d**) showing the distribution of results obtained for the commercial ELISAs and the SU5 ELISA. Exp. Positive/Exp. Negative: Expected positive (blue dots) and negative (red dots) results were defined according to the composite reference standard. Sens, Spec, and PPV: sensitivity, specificity and positive predictive value. Pre-specified assay thresholds are shown as red lines for **a**-**c**. Agrmt indicates the agreement between the real-time PCR results and the SU5 ELISA.

**Figure 3 pathogens-11-00129-f003:**
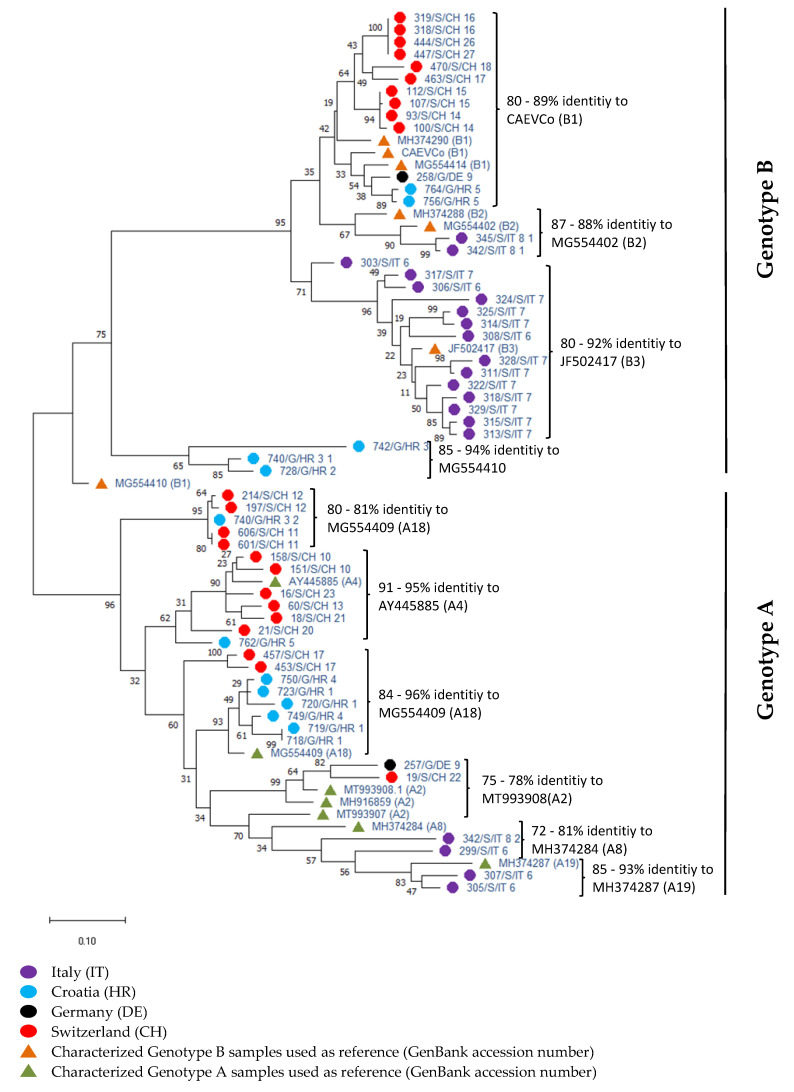
Phylogenetic tree constructed using a 200-bp LTR-gag region located within the target sequence of the real-time PCR. The tree was constructed using the maximum likelihood method using the Tamura–Nei substitution model. A total of 57 sequences of 2 German, 12 Croatian, 20 Italian and 23 Swiss samples of this study were included in the analysis. Moreover, 14 closely related sequences from Genbank were used as a reference. The sample names contain information regarding the lab ID, species (G = goat and S = sheep), country of origin (HR = Croatia, IT = Italy, DE = Germany, CH = Switzerland) and the continuous numbering of flocks tested. The genotypes groups A and B correspond to MVV and CAEV, respectively, in classical terminology.

**Figure 4 pathogens-11-00129-f004:**
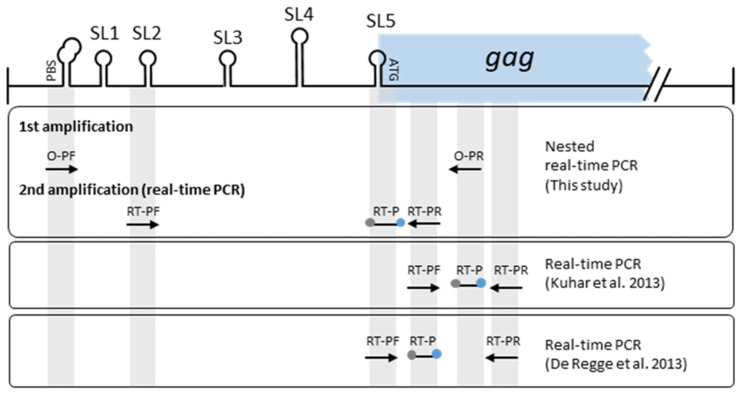
Schematic diagram showing the SRLV target regions of primers and probes. Abbreviations: O-PF: outer primer forward; O-PR: outer primer reverse; RT-PF: real-time PCR primer forward; RT-PR: real-time PCR primer reverse; RT-P: real-time PCR fluorescent probes; SL: stem loop structure (according to Bjarnadottir et al. 2006).

**Table 1 pathogens-11-00129-t001:** Heatmap for the serological and the real-time PCR results of all samples.

	Serology														Serology												
				IB * ^5^			SU5 * ^6^					Sero. Results * ^7^	PCR-Results * ^8^					IB * ^5^			SU5 * ^6^					Sero. Results * ^7^	PCR-Results * ^8^
ID Number * ^1^	IDEXX * ^2^	VMRD * ^3^	Eradikit * ^4^	CA	MA	NC	A4	A3	A1	B1	B2			ID Number * ^1^	IDEXX * ^2^	VMRD * ^3^	Eradikit * ^4^	CA	MA	NC	A4	A3	A1	B1	B2		
716/G/HR_1	330	49.6	182	3	1	3	0.33	0.19	0.18	0.22	0.22	S	-	178/S/CH_10	−16	4.4	3	0	0	0	0.01	0.08	0.00	0.11	0.11	-	-
717/G/HR_1	83	7	96	3	0	0	0.16	0.18	0.26	0.18	0.27	S	M	179/S/CH_10	−8	−1.5	2	1	0	1	0.19	0.35	0.42	0.16	0.44	-	-
718/G/HR_1	181	39.1	147	3	0	3	0.46	0.14	0.11	0.14	0.19	M	M	180/S/CH_10	−11	−2.4	3	1	0	1	0.06	0.08	0.21	0.03	0.04	-	-
719/G/HR_1	338	39.4	177	3	2	0	1.02	0.34	0.30	0.38	0.64	M	M	181/S/CH_10	−15	−5.9	14	1	0	1	0.26	0.06	0.00	−0.01	0.05	-	-
720/G/HR_1	87	11.7	46	3	0	1	0.14	0.14	0.15	0.18	0.16	S	M	182/S/CH_10	−16	5.4	39	1	0	0	0.05	0.09	0.08	0.07	0.06	-	-
721/G/HR_1	168	39.9	160	3	0	1	0.16	0.20	0.15	0.18	0.29	S	-	183/S/CH_10	−13	14.4	−1	1	0	0	0.01	0.11	0.02	0.22	0.07	-	-
722/G/HR_1	9	44.6	79	0	0	1	0.12	0.13	0.12	0.14	0.13	S	M	184/S/CH_10	−18	−2	3	1	0	0	−0.06	−0.03	−0.04	0.06	−0.01	-	-
723/G/HR_1	332	50.3	182	3	1	3	0.13	0.14	0.14	0.22	0.14	S	M	185/S/CH_10	−19	−4.7	29	1	0	0	−0.02	−0.03	−0.04	0.03	−0.01	-	-
724/G/HR_1	16	18.1	42	2	0	0	0.61	0.23	0.17	0.33	0.16	-	-	186/S/CH_10	−17	−0.2	1	1	0	0	−0.07	−0.02	−0.03	0.11	0.02	-	-
725/G/HR_1	257	40.3	175	3	1	3	0.13	0.13	0.12	0.28	0.13	S	M	187/S/CH_10	−10	29.5	91	1	0	0	0.43	−0.03	−0.03	−0.12	0.01	-	-
726/G/HR_2	357	87.8	176	3	2	3	3.15	2.21	2.48	4.00	2.52	C	C	188/S/CH_10	−3	34.3		1	0	0	2.68	0.08	0.17	−0.07	0.05	-	-
727/G/HR_2	344	92.5	181	3	2	3	0.52	0.62	0.68	3.11	0.63	C	C	189/S/CH_10	−16	−8.7	3	3	0	0	0.03	0.01	−0.03	−0.03	0.05	-	-
728/G/HR_2	248	90.7	175	3	2	3	0.32	0.54	0.75	3.75	0.28	C	C	190/S/CH_10	19	35.3		0	0	0	2.57	0.12	0.18	0.01	−0.01	-	-
729/G/HR_2	336	77.1	172	3	2	3	0.27	1.80	3.42	4.00	1.15	C	C	191/S/CH_10	209	17.3	216	1	0	0	3.26	2.66	2.00	0.09	0.22	M	-
730/G/HR_2	244	90.7	132	3	2	3	0.17	0.20	0.19	1.53	0.16	C	C	192/G/CH_10	−16	−2.7	12	1	0	0	−0.05	0.00	−0.02	0.08	0.01	-	-
731/G/HR_2	353	89.7	185	3	2	3	0.21	0.35	0.46	1.27	0.25	C	-	193/G/CH_10	−17	−9.9	14	1	0	0	−0.05	0.05	−0.01	0.10	0.05	-	-
732/G/HR_2	302	78	187	3	2	3	0.13	0.52	0.79	4.00	0.18	C	-	596/S/CH_11	79	62.4	113	1	0	1	1.43	2.65	1.56	0.33	0.52	M	M
733/G/HR_2	312	85.4	182	3	2	3	0.14	0.19	0.16	1.52	0.18	C	-	597/S/CH_11	60	57.8	81	0	0	0	1.19	2.38	0.92	0.08	0.14	M	M
734/G/HR_2	341	89.1	177	3	2	3	0.15	0.14	0.14	4.00	0.16	C	C	598/S/CH_11	80	49.3	79	0	0	0	0.39	0.63	0.42	0.30	0.18	M	M
735/G/HR_2	320	83.8	177	3	2	3	0.27	0.64	1.22	1.64	0.12	C	-	599/S/CH_11	73	43.8	127	0	0	0	0.92	1.30	0.41	0.04	0.15	M	M
736/G/HR_3	269	91.7	169	3	1	2	0.71	1.99	2.08	4.00	0.71	C	C	600/S/CH_11	354	69.5	149	0	0	0	0.95	2.32	1.92	1.45	0.06	M	M
737/G/HR_3	211	90.5	149	3	1	2	0.18	0.35	3.65	1.04	0.30	C	-	601/S/CH_11	283	46.2	141	0	0	0	0.36	1.16	1.40	0.07	0.15	M	M
738/G/HR_3	200	83.3	183	3	2	3	0.26	0.29	0.17	3.65	0.14	C	-	602/S/CH_11	155	46.4	169	0	0	0	0.36	0.32	0.18	−0.02	−0.04	M	M
739/G/HR_3	217	89.2	128	3	0	1	0.20	0.19	0.24	0.36	0.16	C	C	603/S/CH_11	140	74.2	142	0	0	0	3.04	0.82	0.32	0.46	0.20	M	M
740/G/HR_3	352	72.1	171	3	2	3	0.11	0.48	0.60	3.32	0.11	C	MC	604/S/CH_11	106	77.7	126	0	0	0	3.09	1.80	0.97	0.06	0.04	M	M
741/G/HR_3	292	92.5	162	3	2	3	1.81	1.47	2.47	4.00	2.29	C	C	605/S/CH_11	113	51	135	0	0	0	1.67	0.74	0.54	−0.01	0.00	M	M
742/G/HR_3	208	89.6	129	3	1	1	0.61	0.50	1.08	2.92	0.56	C	C	606/S/CH_11	149	55.9	143	0	0	0	1.92	0.86	0.60	0.04	0.23	M	M
743/G/HR_3	346	91.6	160	3	2	3	0.59	0.45	0.63	4.00	0.42	C	C	607/S/CH_11	−3	1.3	−2	0	0	0	0.08	−0.01	0.01	0.00	0.05	-	-
744/G/HR_3	291	93.8	134	3	2	3	0.30	0.27	0.41	3.85	0.45	C	C	608/S/CH_11	79	31.5	50	0	0	0	0.02	0.07	0.00	−0.01	0.03	S	-
745/G/HR_3	353	92.7	162	3	1	3	0.74	0.85	0.81	4.00	0.75	C	C	609/S/CH_11	293	67.4	155	0	0	0	0.87	0.56	1.14	0.08	0.07	M	M
746/G/HR_4	−9	11.4	−3	0	0	0	0.14	0.13	0.10	0.14	0.12	-	-	610/S/CH_11	233	71.2	159	0	0	0	0.31	0.50	0.14	0.00	0.27	M	M
747/G/HR_4	116	48	148	3	1	3	0.19	0.14	0.14	0.16	0.14	S	-	611/S/CH_11	173	60.7	156	0	0	0	0.33	0.65	0.12	0.05	0.01	M	M
748/G/HR_4	88	36.8	98	3	0	1	0.24	0.35	0.22	0.39	0.41	C	C	612/S/CH_11	232	45.6	127	0	0	0	0.89	2.03	0.62	0.02	0.60	M	M
749/G/HR_4	228	60.5	152	2	1	3	0.44	0.22	0.12	0.18	0.13	M	M	613/S/CH_11	145	35.7	105	0	0	0	0.14	1.17	0.09	−0.01	0.14	M	M
750/G/HR_4	141	39.4	101	3	0	1	0.13	0.25	0.11	0.17	0.14	S	M	614/S/CH_11	240	31.3	90	0	0	0	0.17	0.31	0.08	−0.01	0.00	S	M
751/G/HR_4	−9	5.8	0	0	0	0	0.30	0.41	0.12	0.19	0.17	-	-	615/S/CH_11	6	17.7	3	0	0	0	0.00	−0.02	0.01	−0.02	0.01	-	-
752/G/HR_4	−1	10.4	0	0	0	1	0.22	0.15	0.11	0.14	0.12	-	-	616/S/CH_11	9	37.3	9	0	0	0	−0.04	−0.03	−0.01	0.09	0.08	-	-
753/G/HR_4	243	79.1	119	3	0	3	0.16	0.15	0.43	0.14	0.17	M	-	617/S/CH_11	121	47.8	84	0	0	0	0.00	0.60	0.05	0.01	0.10	M	M
754/G/HR_4	−5	−1.3	5	1	0	0	0.14	0.19	0.14	0.22	0.22	-	-	618/S/CH_11	108	35	−3	2	0	0	0.06	−0.07	0.02	0.02	0.30	S	M
755/G/HR_4	248	31.6	129	3	2	3	0.14	0.14	0.11	0.13	0.14	S	-	619/S/CH_11	2	13.9	2	0	0	0	−0.01	0.00	0.00	−0.02	−0.03	-	-
756/G/HR_5	352	81.9	150	3	2	3	0.21	0.22	0.21	4.00	0.13	C	C	620/S/CH_11	5	16.2	3	1	0	0	0.00	0.17	0.03	0.01	0.02	-	-
757/G/HR_5	306	92.8	128	3	2	3	0.27	0.48	0.23	1.24	0.51	C	C	621/S/CH_11	4	26.8	9	0	0	0	0.04	0.05	0.06	−0.01	0.02	-	-
758/G/HR_5	200	87.8	157	3	2	3	0.95	1.55	1.16	4.00	0.26	C	C	622/S/CH_11	22	51.1	18	0	0	0	0.01	0.31	0.11	0.00	0.05	-	-
759/G/HR_5	278	89.1	141	3	2	3	0.12	0.16	0.13	2.14	0.16	C	C	623/S/CH_11	272	53	36	2	0	0	−0.01	0.20	0.05	0.00	0.13	S	M
760/G/HR_5	300	77.4	152	3	2	3	0.15	0.20	0.24	1.35	0.19	C	-	624/S/CH_11	22	41	38	1	0	0	−0.04	−0.09	−0.01	−0.03	0.01	-	-
761/G/HR_5	231	76.9	136	3	2	3	0.20	0.20	0.17	0.43	0.28	C	C	625/S/CH_11	20	25.9	36	1	0	0	0.11	−0.06	0.00	0.01	0.07	-	-
762/G/HR_5	119	50	115	3	1	1	0.12	0.15	0.17	0.38	0.12	C	M	626/S/CH_11	21	22	65	1	0	0	−0.01	−0.04	0.01	−0.02	0.00	-	-
763/G/HR_5	189	46.3	53	3	0	0	0.32	0.12	0.12	0.37	0.11	C	C	627/S/CH_11	13	30	52	0	0	0	0.26	0.17	0.07	0.00	0.04	-	-
764/G/HR_5	155	88.8	130	3	1	2	0.49	1.19	1.39	3.18	0.63	C	C	628/S/CH_11	16	36.2	49	1	0	0	0.52	0.48	0.20	0.06	0.08	M	-
765/G/HR_5	212	46.4	112	3	1	2	0.14	0.13	0.12	3.82	0.15	C	C	629/S/CH_11	268	56.7		2	0	0	0.98	1.08	1.29	0.07	0.07	M	M
299/S/IT_6	305	81	105	1	0	1	0.40	0.16	0.18	0.21	0.28	M	M	630/S/CH_11	41	23.1	49	1	0	0	−0.01	0.13	0.06	−0.02	0.50	M	M
300/S/IT_6	269	92.7	118	1	0	1	1.96	2.51	1.65	2.42	2.34	M	C	631/S/CH_11	10	7.2	4	1	0	0	0.02	0.34	0.03	0.12	0.38	-	-
301/S/IT_6	400	91.3	122	1	0	1	0.81	2.20	0.68	1.82	0.96	M	C	194/S/CH_12	−15	8.6	1	1	0	0	0.26	0.06	0.00	0.07	0.05	-	-
302/S/IT_6	304	41.2	47	1	0	0	1.56	0.57	0.33	1.85	1.57	C	C	195/S/CH_12	−9	−3.2	1	0	0	0	−0.03	0.03	−0.01	−0.07	0.07	-	-
303/S/IT_6	192	94.8	122	1	0	0	2.67	2.28	2.34	2.91	2.14	C	C	196/S/CH_12	−14	−3	0	1	0	0	0.05	0.07	0.97	0.28	0.35	-	-
304/S/IT_6	320	92.4	121	3	2	3	3.96	4.00	3.27	4.00	3.92	C	C	197/S/CH_12	214	64.4	240	1	0	0	3.45	1.48	3.77	0.09	0.73	M	M
305/S/IT_6	219	90.7	105	1	1	1	0.15	0.27	0.18	0.33	1.48	C	MC	198/S/CH_12	−12	1.8	7	1	0	0	0.00	0.07	0.04	0.10	0.20	-	-
306/S/IT_6	205	72.1	117	1	0	2	1.66	1.45	1.40	1.25	0.87	M	MC	199/S/CH_12	8	2.9	15	2	0	0	0.02	0.14	0.25	0.08	0.18	-	-
307/S/IT_6	117	76.1	125	1	0	1	0.12	1.54	0.86	0.05	0.19	M	M	200/S/CH_12	−21	10.6	5	1	0	0	−0.02	0.04	0.02	−0.01	0.10	-	-
308/S/IT_6	295	77.4	114	1	0	3	3.10	3.67	3.90	2.82	2.60	M	MC	201/S/CH_12	160	74.3	179	1	0	0	1.17	2.96	0.69	0.26	0.48	M	M
309/S/IT_6	−7	−17	11	1	0	0	0.19	0.25	0.25	0.20	0.15	-	-	202/S/CH_12	−21	1.5	3	0	0	0	0.36	0.11	0.07	0.17	0.20	-	-
310/S/IT_7	186	37.1	94	2	0	0	0.21	0.26	0.38	0.41	0.34	C	-	203/S/CH_12	−17	6.4	34	1	0	0	−0.02	0.09	0.00	0.15	0.18	-	-
311/S/IT_7	400	83.5	121	2	1	0	1.79	0.32	0.02	0.07	1.56	M	C	204/S/CH_12	80	54.4	211	1	0	0	0.32	0.28	0.15	0.26	0.32	S	M
312/S/IT_7	309	86.9	119	1	0	0	0.33	0.34	0.12	0.22	0.28	S	-	205/S/CH_12	66	57.1	147	1	0	0	1.26	0.66	0.01	0.09	0.11	M	M
313/S/IT_7	158	50.8	54	1	0	0	1.12	0.34	0.92	0.04	0.01	M	C	206/S/CH_12	−19	3.2	6	1	0	0	0.03	0.23	0.09	0.27	0.31	-	-
314/S/IT_7	193	92.6	126	0	0	0	0.06	0.41	0.11	0.13	0.10	M	C	207/S/CH_12	217	69.3	221	1	0	0	0.08	0.04	0.28	0.07	0.05	S	M
315/S/IT_7	400	84.1	125	3	0	0	0.31	0.77	0.97	0.50	0.05	M	C	208/S/CH_12	−20	8.9	1	1	0	0	0.43	0.29	2.48	0.40	0.32	-	-
316/S/IT_7	280	80	124	3	0	0	0.14	0.35	0.59	0.14	0.14	M	C	209/S/CH_12	194	68	235	2	0	0	2.89	2.57	1.88	0.14	0.25	M	M
317/S/IT_7	340	87.4	118	2	0	0	1.60	2.41	2.17	0.45	1.33	M	C	210/S/CH_12	197	68	250	1	0	2	4.00	4.00	4.00	0.14	0.15	M	M
318/S/IT_7	108	90.3	85	2	0	0	0.39	0.65	0.66	0.50	0.31	M	C	211/S/CH_12	48	45	229	1	0	0	0.84	2.46	2.49	0.06	0.30	M	M
319/S/IT_7	174	58.7	93	1	0	0	0.10	−0.10	0.07	0.04	0.06	S	-	212/S/CH_12	103	66.3	252	1	0	0	2.73	1.36	3.27	0.15	0.22	M	M
320/S/IT_7	100	80.4	126	1	0	0	0.35	0.57	0.02	0.07	0.10	M	-	213/S/CH_12	64	49.2	257	1	0	0	0.33	0.45	0.89	0.14	0.23	M	-
321/S/IT_7	400	87.8	123	3	0	0	0.05	1.26	1.26	0.26	0.08	M	C	214/S/CH_12	36	44.8	220	0	0	0	0.97	0.46	0.69	0.10	0.31	M	M
322/S/IT_7	302	73.3	121	2	2	2	0.47	0.82	0.42	0.47	0.68	M	C	215/S/CH_12	212	67.5	226	1	0	0	0.69	1.41	1.91	0.05	0.10	M	M
323/S/IT_7	−5	−24	7	1	0	0	0.21	0.10	0.06	0.25	0.12	-	-	216/S/CH_12	8	28.8	35	0	0	0	0.14	0.22	0.07	−0.01	0.06	-	M
324/S/IT_7	324	66.7	97	3	1	1	0.30	0.45	0.18	0.73	0.22	C	C	217/S/CH_12	−3	41.6	81	0	0	0	0.15	1.45	0.02	0.03	0.17	M	-
325/S/IT_7	295	82.7	115	2	1	1	0.25	0.24	0.18	0.12	0.13	S	C	218/S/CH_12	1	46.1	145	1	0	0	0.04	0.06	0.05	0.10	−0.02	S	-
326/S/IT_7	−6	−18	15	1	0	0	0.09	−0.06	0.10	0.12	0.08	-	-	219/S/CH_12	−16	20.3	8	0	0	0	0.19	−0.01	0.02	0.07	−0.03	-	-
327/S/IT_7	−10	−19	81	0	0	0	0.01	0.06	0.02	0.05	0.01	-	-	220/S/CH_12	17	55.8	170	1	0	0	0.00	0.21	0.07	0.18	0.05	S	-
328/S/IT_7	346	72.2	131	3	0	0	1.67	0.19	0.13	0.39	0.48	M	C	221/S/CH_12	5	52.8	125	1	0	0	0.16	0.13	−0.01	0.03	−0.02	S	-
329/S/IT_7	186	87.9	121	1	0	0	0.36	0.56	0.36	0.57	0.57	C	C	222/S/CH_12	16	25.3	194	1	0	0	0.47	0.22	0.25	0.21	0.22	-	-
330/S/IT_8	−13	−9	−1	1	0	0	0.29	0.61	0.23	0.07	0.03	-	-	60/S/CH_13	158	70.1	173	2	0	2	3.00	1.87	3.50	0.21	0.20	M	M
331/S/IT_8	57	49.9	115	1	0	0	0.82	0.51	0.59	0.21	0.60	M	-	62/S/CH_13	125	15.5	85	1	0	0	3.89	3.17	3.25	2.05	0.01	M	M
332/S/IT_8	−8	−11	28	1	0	0	0.06	0.15	0.04	0.05	0.04	-	-	64/S/CH_13	109	30.3	145	2	0	0	4.16	3.57	4.15	0.03	0.04	M	M
333/S/IT_8	204	59.7	121	2	0	1	0.11	0.70	0.27	0.33	0.07	M	M	65/S/CH_13	107	21.8	87	2	0	0	4.25	3.90	4.21	0.16	1.23	M	-
334/S/IT_8	86	49.2	113	1	0	1	0.23	0.00	0.13	1.73	0.11	C	M	68/S/CH_13	75	15.4	32	0	0	0	3.86	1.90	1.98	1.32	1.28	-	-
335/S/IT_8	−8	−11	−1	0	0	0	0.13	−0.08	0.07	0.22	0.05	-	-	70/S/CH_13	92	56.2	24	1	0	0	3.11	2.90	2.45	0.25	0.26	M	M
336/S/IT_8	−12	−6.1	−4	0	0	0	0.07	0.21	0.09	0.07	0.07	-	-	93/S/CH_14	372	76.1	92	2	1	3	2.01	2.25	2.01	1.55	1.78	M	C
337/S/IT_8	−16	−11	−4	0	0	0	0.07	−0.07	0.03	0.04	0.01	-	-	94/S/CH_14	185	69.1	24	1	0	0	0.80	−0.13	−0.03	−0.04	0.99	M	-
338/S/IT_8	−6	−15	−1	0	0	0	0.08	0.35	0.11	0.07	0.05	-	-	95/S/CH_14	56	81.9	28	1	0	0	0.04	−0.04	−0.03	−0.08	−0.06	S	C
339/S/IT_8	130	56.2	117	1	0	0	1.02	0.17	1.03	0.26	0.99	M	M	99/S/CH_14	349	61.9	65	1	0	0	0.38	0.12	0.13	−0.09	0.88	C	-
340/S/IT_8	−6	2	48	0	0	0	0.97	0.19	0.79	0.15	0.12	-	M	100/S/CH_14	420	90	113	1	0	2	0.86	0.02	1.76	0.14	0.27	M	C
341/S/IT_8	−15	−8.4	10	0	0	0	0.40	0.22	0.05	0.26	0.07	-	-	101/S/CH_14	359	66.4	101	1	0	0	0.05	−0.18	0.07	−0.04	0.46	C	-
342/S/IT_8	93	80.2	111	1	0	1	0.97	1.28	1.11	2.64	0.54	C	MC	102/S/CH_14	128	70.1	2	1	0	0	−0.02	0.37	−0.03	−0.10	0.34	M	-
343/S/IT_8	163	94.4	117	1	0	1	0.00	0.06	0.11	0.81	−0.02	C	C	103/S/CH_14	231	60.6	30	1	0	0	0.92	−0.18	−0.05	0.00	0.28	M	C
344/S/IT_8	154	60.2	111	2	0	1	1.02	1.56	2.17	2.23	0.00	C	-	105/S/CH_15	195	87.7	11	0	0	0	2.40	−0.15	−0.05	−0.10	1.58	M	-
345/S/IT_8	309	87.3	124	3	0	1	0.66	0.27	1.49	2.62	0.40	C	MC	106/S/CH_15	142	86.2	36	1	0	0	0.34	−0.20	−0.06	−0.10	0.83	C	-
346/S/IT_8	−22	−6.2	−1	0	0	0	0.02	0.02	0.04	0.03	0.01	-	-	107/S/CH_15	456	65.3	86	3	1	2	0.02	−0.01	−0.04	0.62	−0.10	C	C
347/S/IT_8	40	30.7	94	2	0	0	3.04	0.20	4.00	0.40	0.42	M	-	108/S/CH_15	27	74	16	0	0	0	0.37	−0.17	−0.04	−0.09	0.21	-	C
348/S/IT_8	−9	23.4	34	0	0	0	0.82	0.46	0.22	0.20	0.40	-	-	109/S/CH_15	123	87.5	7	1	0	0	−0.09	−0.19	0.04	−0.10	0.12	S	-
349/S/IT_8	−10	−5.7	0	0	0	0	0.35	0.60	0.21	0.26	0.09	-	-	110/S/CH_15	184	82.5	45	1	0	1	−0.01	0.13	−0.05	0.00	−0.04	S	-
255/G/DE_9	−17	−82	30	0	0	0	0.03	−0.01	0.12	−0.03	−0.03	-	-	111/S/CH_15	167	68.5	46	1	0	1	1.18	−0.16	−0.04	−0.07	1.04	M	-
256/G/DE_9	−18	−80	88	0	0	0	0.04	0.00	−0.02	−0.02	−0.01	-	-	112/S/CH_15	112	79	17	1	0	0	−0.12	−0.19	−0.08	−0.05	0.14	S	C
257/G/DE_9	246	49	96	0	0	0	0.02	0.39	−0.01	0.00	−0.04	M	M	316/S/CH_16	184	80.3	31	2	0	0	0.01	−0.08	0.08	0.35	0.03	C	C
258/G/DE_9	348	71.5	124	3	0	0	0.00	0.05	0.01	0.23	−0.05	S	C	318/S/CH_16	257	79.6	44	3	1	3	0.06	0.07	0.07	2.17	0.17	C	C
259/G/DE_9	−17	−73	−4	0	0	0	−0.02	0.05	0.00	0.07	−0.02	-	-	319/S/CH_16	257	71.6	75	3	2	3	0.22	0.02	0.06	2.21	0.22	C	C
260/G/DE_9	−18	−43	−4	0	0	0	−0.01	0.06	0.04	0.15	0.05	-	-	320/S/CH_16	238	77.8	25	3	1	3	0.00	0.54	0.10	0.16	0.03	M	C
261/G/DE_9	−18	−54	−7	0	0	0	0.00	−0.02	0.26	−0.01	−0.06	-	-	321/S/CH_16	215	80.8	174	3	2	3	1.21	2.04	1.73	2.34	2.03	C	-
262/G/DE_9	−13	−54	−7	0	0	0	−0.01	−0.02	0.14	0.02	−0.03	-	-	322/S/CH_16	253	85.2	16	3	2	0	0.05	0.16	0.14	0.02	0.03	S	C
263/G/DE_9	−19	−53	−7	0	0	0	0.00	0.02	0.08	0.04	0.02	-	-	323/S/CH_16	93	76.2	83	0	1	1	0.02	−0.03	−0.01	0.82	0.23	C	C
264/G/DE_9	−11	−47	−6	0	0	0	0.03	0.00	0.01	−0.03	−0.05	-	-	324/S/CH_16	222	87.2	81	3	0	0	0.00	−0.06	0.01	0.04	0.01	S	C
265/G/DE_9	−15	−64	−3	0	0	0	0.00	0.04	−0.01	0.01	−0.04	-	-	325/S/CH_16	210	66.1	28	3	1	3	−0.02	0.02	0.02	0.72	0.68	C	C
266/G/DE_9	−18	−38	−7	0	0	0	0.01	0.07	−0.02	0.21	0.06	-	-	326/S/CH_16	135	82.1	115	2	0	0	−0.02	0.00	0.04	0.69	0.03	C	C
267/G/DE_9	−15	−31	−5	0	0	0	0.10	0.06	0.13	0.07	0.04	-	-	327/S/CH_16	250	85.6	56	3	2	3	0.29	0.49	0.40	0.12	0.24	M	C
268/G/DE_9	−17	−58	−7	0	0	0	0.20	0.29	−0.04	0.02	−0.06	-	-	328/S/CH_16	3	67.4	6	1	0	0	−0.02	0.05	0.03	0.03	0.06	-	-
269/G/DE_9	−18	−27	−6	0	0	0	0.01	0.30	−0.01	−0.01	−0.05	-	-	452/S/CH_17	45	35.5	40	1	0	0	0.20	0.14	0.13	0.08	0.21	S	M
270/G/DE_9	−15	−27	−8	0	0	0	0.00	0.01	−0.03	−0.02	−0.08	-	-	453/S/CH_17	55	45.9	37	1	0	0	0.11	0.19	0.62	−0.01	0.23	M	M
271/G/DE_9	−18	−29	2	0	0	0	0.00	0.09	−0.03	0.08	0.01	-	-	455/S/CH_17	81	39	16	1	0	1	1.17	0.03	0.04	0.12	0.02	M	-
151/S/CH_10	37	41	118	0	0	0	2.34	0.01	0.13	0.04	0.14	M	M	456/S/CH_17	68	12.1	29	nd	nd	nd	1.28	1.48	0.52	0.21	0.16	-	M
152/S/CH_10	−20	−0.4	13	0	0	0	−0.02	0.04	0.00	0.16	0.09	-	-	457/S/CH_17	31	53.1	31	1	0	0	0.08	−0.01	0.04	−0.02	0.13	-	M
153/S/CH_10	44	71.8	76	0	0	0	2.10	0.53	1.12	0.16	0.35	M	-	458/S/CH_17	13	37.2	6	1	0	0	0.02	0.10	0.03	0.01	0.15	-	M
154/S/CH_10	36	33.5	10	0	0	0	0.73	−0.01	0.05	−0.09	0.00	-	-	459/S/CH_17	154	30.7	51	2	0	0	2.36	2.18	2.33	1.52	2.97	C	M
155/S/CH_10	21	40	161	0	0	0	2.17	0.06	0.21	−0.01	0.14	M	M	461/S/CH_17	250	84.7	75	3	2	3	0.16	−0.05	0.01	−0.10	4.05	C	-
156/S/CH_10	−13	1.9	2	0	0	0	0.15	0.12	0.01	−0.02	0.07	-	-	462/S/CH_17	167	42.4	0	3	0	0	−0.02	−0.08	0.02	0.04	0.06	S	-
157/S/CH_10	67	37.5	26	0	0	0	1.34	0.61	0.37	0.56	0.36	M	-	463/S/CH_17	205	82.6	49	3	2	3	0.10	0.14	0.31	0.00	0.32	S	C
158/S/CH_10	229	70.2	234	0	0	0	4.00	0.31	2.06	0.08	0.78	M	M	465/S/CH_18	250	62	60	3	3	3	3.77	−0.13	1.53	−0.03	3.69	M	C
159/S/CH_10	32	45.2	47	0	0	0	2.89	0.03	0.10	0.07	0.07	M	-	467/S/CH_18	125	36.1	7	2	0	0	0.04	−0.05	−0.01	0.03	0.84	C	C
160/S/CH_10	3	1		0	0	0	0.09	0.51	0.09	0.87	0.53	-	-	468/S/CH_18	79	26.4	0	2	0	0	0.39	−0.10	0.07	−0.08	0.28	-	C
161/S/CH_10	−13	−4.4	20	0	0	0	0.06	0.16	0.05	0.29	0.19	-	-	469/S/CH_18	157	59.9	151	2	0	1	0.83	0.17	0.23	−0.02	2.56	C	C
162/S/CH_10	377	38.4	233	3	0	0	4.00	3.75	3.46	0.16	0.71	M	M	470/S/CH_18	215	44.2	74	3	2	3	0.32	−0.09	0.13	0.03	0.58	C	C
163/S/CH_10	−17	4.8	60	1	0	0	0.01	0.04	0.05	−0.07	0.08	-	-	471/S/CH_18	136	87.8	70	2	2	0	0.65	0.07	0.05	0.74	0.51	C	C
164/S/CH_10	−14	3.7	0	1	0	0	0.59	0.42	0.05	0.31	0.30	-	-	472/S/CH_18	47	45.1	22	2	0	1	0.19	−0.09	−0.01	0.07	0.26	S	C
165/S/CH_10	−14	−3.4	5	2	0	0	0.16	0.14	0.00	0.22	0.12	-	-	473/S/CH_18	199	90.5	62	2	2	3	0.88	0.91	0.55	0.24	0.81	M	C
166/S/CH_10	54	48.6	223	0	0	0	4.00	0.49	0.76	0.09	0.07	M	-	123/S/CH_19	−13	10	nd	0	0	0	0.08	0.17	0.12	0.12	0.03	-	-
167/S/CH_10	−18	3.8	5	0	0	0	0.06	0.07	0.02	−0.01	0.06	-	-	124/S/CH_19	153	26	nd	2	1	1	3.57	4.00	2.26	2.90	1.00	-	M
168/S/CH_10	−12	3.7	7	0	0	0	0.10	0.30	0.33	0.17	0.22	-	-	21/S/CH_20	121	25	109	1	0	0	1.20	0.61	0.46	0.12	0.14	M	M
169/S/CH_10	−11	−3.3	2	0	0	0	−0.02	0.13	0.00	0.10	0.14	-	-	18/S/CH_21	48	58	76	1	0	0	1.65	1.78	1.33	0.64	1.17	M	M
170/S/CH_10	−14	0.5	1	0	0	0	−0.04	0.07	0.01	0.09	0.06	-	-	19/S/CH_22	99	53	74	1	0	0	2.53	0.26	2.42	1.07	0.71	M	M
171/S/CH_10	22	10.2	6	0	0	0	1.11	0.05	0.08	−0.06	0.06	-	-	16/S/CH_23	149	14	23	1	0	1	2.71	0.00	0.09	0.01	0.06	-	M
172/S/CH_10	−16	5.8	9	1	0	0	0.40	0.14	−0.02	0.03	0.12	-	-	15/S/CH_23	63	25	74	1	0	0	2.52	0.68	0.99	0.10	0.40	M	M
173/S/CH_10	−2	−3.3	14	0	0	0	−0.03	0.07	−0.01	0.16	0.06	-	-	82/S/CH_24	94	40.8	105	1	0	0	2.99	0.93	0.69	0.83	0.66	M	-
174/S/CH_10	−6	2	7	0	0	0	0.36	0.23	0.39	0.66	0.22	-	-	78/S/CH_25	203	64.6	44	2	0	3	3.99	4.13	3.96	1.44	0.12	M	-
175/S/CH_10	−18	−0.1	3	1	0	0	0.08	0.02	0.01	−0.02	0.02	-	-	444/S/CH_26	184	82.9	71	3	1	3	−0.02	−0.10	−0.01	−0.12	−0.01	S	C
176/S/CH_10	−8	12.2	0	1	0	0	0.07	0.14	0.12	0.25	0.18	-	-	445/S/CH_26	−2	4	0	0	0	0	0.05	−0.05	0.03	−0.06	0.01	-	-
177/S/CH_10	−14	−7.6	4	0	0	0	0.01	0.14	0.09	0.10	0.12	-	-	447/S/CH_27	250	81.7	56	3	1	3	0.27	0.53	0.44	0.38	0.54	C	C

^1^ ID number consists of lab number, species (G = goat and S = Sheep), country (HR = Croatia, IT = Italy, DE = Germany, CH = Switzerland) and the continuous numbering of flocks tested; ^2^ results of IDEXX ELISA, corrected OD; ^3^ results of VMRD ELISA, percent of inhibition; ^4^ results of ERADIKIT ELISA, corrected OD; ^5^ Immunoblot, CA (capsid, p25), MA (matrix, p18) and NC (nucleocapisd, p15), 0 = no reaction, 1 = slight reaction, 2 = moderate reaction, 3 = strong reaction; ^6^ SU5 ELISA, corrected OD. ^7^ **Serological results:** S (gray cell background) = SRLV if SU5 is negative but 2 of the commercial ELISAs are positive; M (green cell background) = MVV if SU5 positive at A4, A3, A1; C (orange cell background) = CAEV if SU5 is positive at B1 or B2. ^8^ **PCR results**: M for MVV (green cell background) = RT-PCR MVV-positive; C for CAEV (orange cell background) = RT-PCR CAE-positive; MC (yellow cell background) = RT-PCR MVV- and CAE-positive. The colors reflect the intensity of each serological reaction, with values ranging from dark to light blue to white (negative) and from light to dark red (positive).

**Table 2 pathogens-11-00129-t002:** Diagnostic performance of the screening ELISAs, Immunoblot and RT-PCR crossing the results of each test with the composite truth standard.

	Sensitivity		Specificity		Pos. Predictive Value		Neg. Predictive Value	
	Value %	95% CI % ^1^	Value %	95% CI % ^1^	Value %	95% CI % ^1^	Value %	95% CI % ^1^
IDEXX ELISA	92.2	87.4–95.6	98.9	94.1–100	99.4	96.2–99.9	85.9	77.7–91.9
VMRD ELISA	90.1	85.0–93.9	95.7	89.2–98.8	97.7	94.3–99.1	82.2	73.7–89.0
ERADIKIT ELISA	84.4	78.5–89.2	91.3	83.6–96.1	95.3	91.2–97.5	73.7	64.6–81.5
Immunoblot	59.2	51.8–66.2	92.4	85.0–96.9	94.2	88.4–97.6	52.2	44.2–60.0
Real-time PCR	75.5	68.8–81.4	100	96.1–100	100	97.5–100	66.2	57.7–74.0

^1^ 95% CI % = 95% Confidence Interval in %.

**Table 3 pathogens-11-00129-t003:** Diagnostic performance of the immunoblot test and the nested real-time PCR.

Immunoblot					Nested Real-Time PCR				
Truth	Pos	Neg	Total	Percent	Truth	Pos	Neg	Total	Percent
Pos	113	78	191	59.2 (Se)	Pos	145	47	192	75.5 (Se)
Neg	7	85	92	92.4 (Sp)	Neg	0	92	92	100 (Sp)
Total	120	163	283	100	Total	145	139	284	100
Percent	94.2 (P)	52.1 (N)	100		Percent	100 (P)	66.2 (N)	100	

Se = sensitivity, Sp = specificity, P = positive predictive value; N = negative predictive value.

**Table 4 pathogens-11-00129-t004:** Immunoblot failure by animal host.

	I-Blot pos		I-Blot neg		
	MVV^a^ (%)	CAEV^a^ (%)	MVV^a^ (%)	CAEV^a^ (%)	N (% Detected)
**Sheep**	14 (26.9)	38 (73.1)	43 (82.7)	9 (17.3)	104 (50.0)
**Goats**	9 (28.1)	23 (71.9)	2 (100)	0 (0)	34 (94.1)
**Total**	23 (16.7)	45 (32.6)	61 (44.2)	9 (6.5)	138 (49.3)

Viral species assigned by nested real-time PCR.

**Table 5 pathogens-11-00129-t005:** Immunoblot failure by viral species.

	I-Blot pos		I-Blot neg		
	Sheep	Goats	Sheep	Goats	N (% Detected)
**MVV^a^**	14 (60.9)	9 (39.1)	43 (95.6)	2 (4.4)	68 (33.8)
**CAEV^a^**	38 (62.3)	23 (37.7)	9 (100)	0 (0)	70 (87.1)
**Total**	52 (37.7)	32 (23.2)	52 (37.7)	2 (1.4)	138 (60.9)

Viral species assigned by nested real-time PCR.

**Table 6 pathogens-11-00129-t006:** Samples tested in this study listed by country, number of sheep, goats and farms.

Country	Number of Sheep	Number of Goats	Number of Farms
Croatia	0	50	5
Italy	51	0	3
Germany	0	17	1
Switzerland	170	2	18

**Table 7 pathogens-11-00129-t007:** Sequences of primers and probes for the nested real-time PCR system. IUPAC codes were used to indicate degenerated primers. ^a^ F indicates forward primer, R reverse primer and P fluorescent probe, respectively, ^b^ GenBank accession number M33677 used as a reference sequence to indicate positions of primers and probes’ binding sites, and ^c^ GenBank accession number used as a reference sequence.

Nested Real-Time PCR		Oligonucleotide ^a^	Primer Sequence/Flouorescent Dye	Position
**First-step PCR:**	outer primers	Outer SRLV-F1	5′-CGCAGGTGGCGCCCAG-3′	158–173 ^b^
		Outer SRLV-F2	5′-CGCAGSTGGCGCCCAA-3′	158–173 ^b^
		Outer SRLV-R1	5′-CCTTCTGTCAGGCGCTCCCC-3′	622–642 ^b^
		Outer SRLV-R2	5′-CCTTCCGTCAAGGTCTCCTTCC-3′	621–642 ^b^
		Outer SRLV-R3	5′-CCTTCTGTCAAGGTCTCCTTCCC-3′	620–642 ^b^
		Outer SRLV-R4	5′-CCTTCTGTCAAGTGCTCCCCTCT-3′	620–642 ^b^
		Outer SRLV-R5	5′-CCTTCTGTCAGGTGCTCCCCTCT-3′	620–642 ^b^
**Second-step:**	Genotype Aspecific real-time PCR	RTMVVLTRgag-F	5′-GGGGACGCCTGAAGTRAGGTAA-3′	287–308 ^c^
		RTMVVLTRgag-R	5′-YTTGAGCTCRGGGTAYCCCTT-3′	517–537 ^c^
		RTMVVLTRgag-P	5′-FAM-CTTTGAGCCTTGCKTCGCCATGTCT-TAMRA-3′	486–510 ^c^
	Genotype Bspecific real-time PCR	RTCAELTRgag-F	5′-CTGRAGGAGTAMGGTAAGTRACTCTGC-3′	324–350 ^b^
		RTCAELTRgag-R	5′-TTGATRCATTTKTCSAKCTCAGGATAA-3′	365–591 ^b^
		RTCAELTRgag-P	5′-FAM-CCGGAGACTTGCCTCGCCATGTC-TAMRA-3′	530–552 ^b^

## Data Availability

All sequencing data obtained were submitted to NCBI database. Accession numbers are given.

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
