# Peer review of "Evaluation of Serological Methods and a New Real-Time Nested PCR for Small Ruminant Lentiviruses"

_pathogens, 2022, doi:10.3390/pathogens11020129_

Round 1

Reviewer 1 Report

The publication may contribute significantly to improve the extremely complicated SRLV diagnostics. Modern and promising approaches are pointing in the direction of combining serology and PCR diagnostics. However, qPCR methods show a quite limited sensitivity due to the low loads of SRLV infected PBMCs in infected clinically inconspicuous animals. A new broad range real time PCR in a nested format may enhance the sensitivity of the PCR as modern and simple method for detection of SRLV infections.

-Line 75: Is Chekit still the official name of the ELISA test or IDEXX CAEV/MVV Total Ab Test Caprine Arthritis Encephalitis (CAEV)/Maedi-Visna (MVV)?

-It is a little bit difficult to understand the present screening in Switzerland:

Goats: Idexx+VMRD - Sheep: In3diagnostc; The test combination and the downstream cascade will be initialized in case of positive results in the affected animals or in the whole herd?

The same goes for the proposed new screening scheme. Do the authors actually perform even initially the ELISA combination and nested qPCR for all animals in positive herds? (detection of seronegative animals in positive herds was mentioned)

-Why was the ID vet indirect not included; all recent studies describe this ELISA as one of the most sensitive or even the most sensitive tests? Are there any data about the ID vet performance on the sample panel available?

-Do the samples originate from routine diagnostics of the different countries or was there any preselection of problematic or contradictory serological results (like for the samples from Germany).

-Could you explain or discuss the different performance of the serological tests compared to previous studies? The Idexx ELISA systems have been described by several studies (Michiels, Olech) and also by the manufacturer to exhibit a certain weakness in sensitivity compared to other commercial test systems. Could it be explained by preselection or unequal distribution of the samples.

-TABLE 1: Text is barely readable, please try to select another font; orange writing on gray/ orange background is not readable

-Line 195 Did you obtain gag CA sequences of those discordant reacting viruses?

-Figur2 2d

-The authors summarize a weak PCR-SU5 ELISA agreement. But there are substantial differences between sheep and goats that should be explained in the figure (n= xy; differentiated K value?)

-RNAtlys may not be familiar to all readers: “One of the most conserved regions of the lentiviral genome is the RNAtlys primer binding site (PBS-GAACAGGGACUUGAA), where the host lysine transfer RNA hybridizes to the viral RNA genome, serving as a primer for reverse transcription (Ramirez et al., 2013). 

- Table 6 explain the abbreviation b, c

-Line 369. The buffy coats were resuspended in the appropriate volume of buffer xy for one extraction?

Line 380 then

-Data about comparison of the newly developed nested format and published qPCR are completely lacking. The authors conclusively prove the suitability of the PCR resp. the nested PCR products for phylogenetic analyses. However, advantages over the basically includes single-round qPCRs remain unclear. Please include some more characteristics of the nested q PCR: numbers of false positives, limit of detection or lead of positive results over Kuhar or De Regge. The nested PCR format, even in cases of a second round without gel based analyses bear a high risk of obtaining false positive results. Therefore, the advantages should be carefully analysed and weighed.

-Line 449 format?

Author Response

Responses to reviewer 1:

General comment to the Comments conveyed by the Academic Editor:

We have added a remark in the discussion section stating the possibility of higher sensitivity of the PCR (up to 85%) given the quality of samples.

Rev 1, comment 1:

This comment is well taken. We have exchanged the correct name in the whole manuscript.

Rev 1, comment 2:

As added to the introduction section, testing is on the base of individual samples, sampling on individual criteria.

The cascade of testing is triggered by the seropositive result. We have clarified this in the same section as above.

Rev 1, comment 3:

All analyses performed were done on tests/kits established in our laboratory. As far as we have seen in international ring trials, the IDEXX kit is quite brave on our local (molecular) epidemiology. A complete comparison of all available test kits would be very meritorious but is actually far beyond the scope of this study.

Rev 1, comment 4:

Apart from the German samples, samples was not biased from our site as stated in the manuscript and outlined above.

Rev 1, comment 5:

We have addressed this point in comment 2 combined with comment 3.

Rev 1, comment 6:

We stated font colors and cell background more precisely in the legend of the table (lines 110 to 119). Furthermore, the idea of the heatmap is rather to focus qualitatively on the color- than on the character-code. We have added a better explanation of the heatmap as a graphical - rather than numerical tool to visualize the results (lines 95 to 96).

Rev 1, comment 7:

We are aware of the possibility of homologous recombination events. To our knowledge this is a rather rare event in the field of SRLV. Still, we haven’t reached out to this planned sequencing project so far but we have added a statement regarding homologous recombination in the discussion section (lines 307 to 308).

Rev 1, comment 8:

Since PCR results are taken as "true" due to sequencing, agreement is based on absolute numbers, no kappa statistics. Therefore we have adapted the figure 2 accordingly (K value eliminated).

Rev 1, comment 9:

We have based position and sequence of the RNAtlys according to Bjarnadottir et al. and verified its strong conservation over a large panel of sequences in the Geneious package.

Rev 1, comment 10:

Comment is well taken. We had lost the legend in the manuscript indeed. This legend is included now.

Rev 1, comment 11:

As mentioned, the extraction of DNA was performed on 750 microliter EDTA blood, i.e. DNA was finally eluted in a volume of 100 microliter elution buffer. We have added DNA to clarify this (line 398).

Rev 1, comment 12:

Typo, thx! Corrected (line 405)

Rev 1, comment 13:

We are well aware on the risk of contamination in a nested PCR setup. Obviously, the standard procedures to minimize cross contamination were considered. Furthermore, individual amplification products were sequenced at least on a herd basis. Consequently we have considered PCR results as "true", preventing comparative Sens/Spec calculations. Comparisons with published techniques were beyond the scope of this project. Analytical sensitivity was not determined (also addressed in the discussion section in line 316). However, we added a comment in the discussion section regarding the improvement of the detection limit of the nested real-time PCR comparing to the simple real-time PCR using serial dilutions of plasmids containing target sequences based on preliminary results (not shown; line 282).

Rev 1, comment 14:

Corrected, thx.

Reviewer 2 Report

This manuscript presents the results of a large-scale attempt to bring order and clarity into the data testing domestic sheep and goats for the presence of lentiviruses and identifying the viruses as CAEV or MVV. This is a valuable task and the authors have worked hard to identify criteria for validating assay results. However, the manuscript is still, at some places, quite difficult to understand.

As I am at an advanced age, I could not possibly read the labels on Table 1. Some way must be found to use a much larger font.

I was also very puzzled by lines 181-182: “151 out of 184 serologically positive samples could be characterized as MVV or CAEV positive, respectively.” I do not understand this sentence. I think it would have a simple meaning without the “respectively” but I am worried that the “respectively” is supposed to convey some added information, and I do not know what this might be.

Lines 192-193 refer to Figure 3D but I believe 2D was intended.

Line 205 and Figure 3 refer to Genotype A and Genotype B. Is one of these equivalent to CAEV and the other to MVV? If so, please make this explicit; if not, please explain.

Author Response

Responses to reviewer 2:

Rev 2, comment 1:

The idea of the heatmap is rather to focus qualitatively on the color- than on the character-code. We have added a better explanation of the heatmap as a graphical - rather than numerical tool to visualize the results (lines 95 to 96).

Rev 2, comment 2:

We have eliminated the term "respectively" on line 182, which might lead to confusion.

Rev 2, comment 3:

Good point: we have corrected the wrong number accordingly (line 194 to 195).

Rev 2, comment 4:

This point is well taken, we have added this important assignment to the figure legend (line 305 and 306).